# Modulation of Virulence Factors during *Trypanosoma cruzi* Differentiation

**DOI:** 10.3390/pathogens12010032

**Published:** 2022-12-25

**Authors:** Camila Oliveira, Fabíola Barbieri Holetz, Lysangela Ronalte Alves, Andréa Rodrigues Ávila

**Affiliations:** 1Laboratório de Regulação da Expressão Gênica, Instituto Carlos Chagas, Fiocruz Paraná, Curitiba 81350-010, Brazil; 2Centre de Recherche CERVO, Université Laval, Québec City, QC G1V 0A6, Canada; 3Research Center in Infectious Diseases, Division of Infectious Disease and Immunity CHU de Quebec Research Center, University Laval, Québec City, QC G1V 4G2, Canada; 4Laboratório de Pesquisa em Apicomplexa, Instituto Carlos Chagas, Fiocruz Paraná, Curitiba 81350-010, Brazil

**Keywords:** gene expression regulation, surface proteins, infective forms, trypanosome, parasite differentiation

## Abstract

Chagas disease is a neglected tropical disease caused by *Trypanosoma cruzi*. This protozoan developed several mechanisms to infect, propagate, and survive in different hosts. The specific expression of proteins is responsible for morphological and metabolic changes in different parasite stages along the parasite life cycle. The virulence strategies at the cellular and molecular levels consist of molecules responsible for mediating resistance mechanisms to oxidative damage, cellular invasion, and immune evasion, performed mainly by surface proteins. Since parasite surface coat remodeling is crucial to invasion and infectivity, surface proteins are essential virulence elements. Understanding the factors involved in these processes improves the knowledge of parasite pathogenesis. Genome sequencing has opened the door to high-throughput technologies, allowing us to obtain a deeper understanding of gene reprogramming along the parasite life cycle and identify critical molecules for survival. This review therefore focuses on proteins regulated during differentiation into infective forms considered virulence factors and addresses the current known mechanisms acting in the modulation of gene expression, emphasizing mRNA signals, regulatory factors, and protein complexes.

## 1. Stage-Specific Proteins That Confer the *Trypanosoma cruzi* Virulence

Chagas disease (American trypanosomiasis) is a disease caused by the protozoan *Trypanosoma cruzi* (family Trypanosomatidae) [1], which includes several species of parasites that cause lethal diseases with a high impact on human health [2,3,4]. It is endemic to the Americas and is on the World Health Organization’s (WHO) list of neglected tropical diseases [5]. The vector pathway remains the main form of *T. cruzi* transmission. However, new forms of infection have emerged, such as blood transfusion, organ transplantation, laboratory accidents, and congenital and oral infection, associated with human migration from the Americas to other continents, making Chagas Disease a global health problem [5]. It is therefore essential to understand the fundamental molecular and cell biology involved in the pathogenesis of this parasite [6]. 

*Trypanosoma cruzi* has a complex life cycle comprising an invertebrate hematophagous triatomine vector (family Reduviidae) and a broad range of mammalian hosts [7]. This parasite has developed distinct mechanisms to infect, propagate, and survive in different environments [8]. Four major developmental stages are identified in both hosts: epimastigotes, blood trypomastigotes, metacyclic trypomastigotes, and amastigotes. The non-infective epimastigotes replicate in the midgut of the insect host and develop into infective metacyclic trypomastigotes in the hindgut (rectum) of the vector, which are released in the excreta during the insect blood feed. The mammalian host can be infected when those infective forms reach the circulatory system through mucous membranes or breaks in the skin. The metacyclic trypomastigotes must invade cells and differentiate into intracellular replicative amastigotes to survive in the mammalian host. After multiplication, the amastigotes differentiate into blood trypomastigotes and are released into the circulatory system to infect new cells [9,10,11]. 

The metacyclogenesis process corresponds to the differentiation of non-infective forms (epimastigotes) into infective forms (metacyclic trypomastigotes) and can be mimicked in vitro [12]. This process is induced by the nutritional stress of epimastigotes which are incubated in a differentiation medium when they adhere to culture flasks before differentiation into metacyclic trypomastigotes [12,13]. For this reason, it is a very suitable model for identifying stage-specific genes and investigating *T. cruzi* differentiation mechanisms [14,15]. The stage-specific gene activation precedes the morphological changes, and the main differences in gene expression are detected within the first 24 h of differentiation when epimastigotes start to express metacyclic-specific antigens [16]. Therefore, in vitro differentiation has become an important method to identify and characterize genes that confer the stage-specific biological properties of infective forms since it is possible to obtain different parasite forms during differentiation into infective forms [17]. 

The use of in vitro metacyclogenesis allowed the initial studies on the transitory expression of stage-specific surface antigens using metabolically labeled proteins and 2D-electrophoresis assays [18]. Obtaining intermediate stages (adhered forms) also allowed both genes to be identified through the representation of differential expression (RDE). These genes are transiently expressed during *T. cruzi* metacyclogenesis and are expressed only in metacyclic trypomastigotes [14]. This approach led to the successful selection of several genes modulated along the metacyclogenesis, although most of them have an unknown function [19]. Those whose function could be ascertained belong to the gene families [20] or putative proteins associated with RNA processing [21].

In general, only a few *T. cruzi* stage-specific genes encoding major surface antigens were initially described before 1999 [22,23,24,25,26]. However, the knowledge of genes involved in pathogenesis, response to environmental changes, and genetic variations among strains increased significantly after sequencing its genome [27]. The first version of the *T. cruzi* genome was published in 2005 using the CLBrenner strain [28] and the improved assemblies using genome sequences of two clones (TcVI and Dm28c) permitted accurate estimation of gene copy number and better annotation of surface multigene families, mucins and trans-sialidases [29]. Currently, there are 38 other genomes of *T. cruzi* strains in databases of the National Center for Biotechnology Information (NCBI) and TriTrypDB [30]. Additionally, the application of next-generation sequencing technologies facilitated the improvement of structural and functional gene annotation and their products in trypanosomatids and provided information about regulatory networks and metabolomes, ushering in a new era of systems biology reviewed elsewhere [31].

Indeed, genome sequencing opened the door to high-throughput technologies, allowing transcriptome and proteome analysis to obtain a systemic view of the gene reprogramming features of the parasite life cycle, which is relevant to identify critical molecules for its survival or even to be used in the control of Chagas disease [32]. 

Methodologies based on proteomic analysis emerged as potential approaches to identify modulated proteins. The first proteomic analysis of *T. cruzi* differentiation was performed in 2004 by comparing three parasite stages using 2DE to identify stage-specific proteins [33]. Further, Atwood et al. [34] used liquid chromatography coupled with a tandem mass spectrometry (LC-MS/MS) approach to identify proteins from different parasite stages. This study described many regulated proteins, highlighting that the differentiation of epimastigotes into metacyclic trypomastigotes resulted in increased expression of proteins involved in antioxidant defenses and decreased ribosomal proteins. The computational image analysis of 2D gels also showed a large proportion of unique proteins expressed during metacyclogenesis, along with evidence that post-translational modifications may be a fundamental part of the parasite’s strategy for regulating gene expression during differentiation [35]. In fact, phosphoproteome analyses by mass spectrometry in a large-scale study showed that the major modulations of phosphorylation sites occur under nutritional stress and after 12 and 24 h of adhesion and regulate proteins involved in the transcription process, transialidases, mucin-associated surface proteins (MASPs), and dispersed gene family 1 (DGF-1) proteins [31].

The employment of complementary techniques also allowed the comparison of the cell surface proteome among the life forms, revealing many membrane proteins involved in host cell infection, protein adhesion, cell signaling, and mammalian host immune response modulation. Several virulence factors, e.g., proteins capable of acting at several host metabolic pathways and regulating the parasite’s cell differentiation, were also found [36,37]. 

In addition to proteomics, methods to evaluate the modulation of genes using analysis of mRNA abundance have been used to identify the genes related to the morphological stage of *T. cruzi* during metacyclogenesis. The first study carried out to understand gene expression in epimastigotes and metacyclic trypomastigotes was performed using microarrays; it revealed that over 50% of genes are regulated, including the transcripts encoding transialidases up-regulated in metacyclic trypomastigotes [38]. The variation in expression profiles between epimastigotes and metacyclic trypomastigotes was further confirmed by SOLiD RNA-seq, evidencing the increase in the expression of transialidases genes related to virulence [39].

Since then, RNA-seq has been used to evaluate transcriptome profiling of different stages. A comparative transcriptomic study among these three main parasite stages revealed the remodeling of surface proteins [40]. This study demonstrated that trypomastigotes exhibit a predominance of surface protein genes, with more than 50% of the transcripts encoding trans-sialidases, MASPs, GP63, mucins, and complement regulatory proteins. In contrast, in amastigotes, the most expressed genes are related to cell cycle, protein, and amino acid catabolic processes, adhesion, and signaling [40]. Another study demonstrated that, during metacyclogenesis, the cellular pathways involved in glucose energy metabolism, amino acid metabolism, and DNA replication are reduced, while processes related to autophagy and cell cycle processes are increased to facilitate the transformation to this infective stage [41].

*Trypanosoma cruzi* has developed virulence strategies at the cellular and molecular levels that englobe molecules responsible for mediating resistance mechanisms to oxidative damage, cellular invasion, and immune evasion [8]. These mechanisms mainly involve parasite surface proteins that allow the parasite to invade diverse cell types and evade the immune system components during the passage through its hosts [42,43]. Comparative transcriptome between two clones of *T. cruzi* strains presenting different virulence patterns confirmed that the differentiation into infective forms is associated with changes in gene expression, mainly in gene families encoding surface proteins, such as trans-sialidases, mucins, and MASPs [44]. 

Thus, parasite surface proteins have emerged as significant virulence factors [6] since the parasite surface coat remodeling is crucial to differentiation processes and infectivity, protecting from host defense mechanisms, in addition to cell attachment and infection [40,45,46,47]. The trypanosome coat is mainly composed of glicophosphatidylinositol (GPI)- anchored glycoconjugates of distinct nature, forming a layer of O-glycosylated mucins and glycoinosilphospholipidis (GIPLs). Glycoproteins, e.g., trans-sialidases (TS), MASPs, GP85, GP63, and others, are also essential components [48]. These glycoproteins belong to multigene families that present a considerable expansion and are identified in the disruptive compartment of the parasite genome [30]. 

TS proteins are critical for interacting with the exogenous environment [49]. They are located on the membrane surface of metacyclic, bloodstream trypomastigotes and are involved in host-parasite interactions [22,50]. The well-known function of these proteins is the trans-sialidase catalytic activity which was first described in 1980 [51]. The parasites cannot synthesize their sialic acids, and they use the TSs to transfer them from the host cell to mucins [52], conferring a negatively charged coat that protects the trypomastigotes from attack by human antibodies [53]. Besides the catalytic function, another study showed that TSs play a crucial role in establishing effective infection since they interact with different human cells [54]. The genes encoding TS or TS-like genes are classified into four groups according to sequence similarity and functional properties. Group I includes members with trans-sialidase activity. Group II consists of proteins involved in host cell attachment and invasion and antibody response, such as Tc85, GP82, and GP90. Group III proteins protect against the complement system, and Group IV consists of the ones with unknown functions [30].

The mucins bear a dense array of oligosaccharides, 0-linked serine, and/or threonine residues and are the acceptors of sialic acids [55,56,57]. They display a high diversity and are classified into two subfamilies: TcMUC (*T. cruzi* mucin genes), which are only expressed in mammalian stages, and TcSMUG (*T. cruzi* small mucin-like genes), which are specific to insect forms [55,58,59]. Some members of TcSMUG are protease-resistant and therefore protect the parasite in the insect vector intestinal tract, while others are implicated in adhesion to the vector midgut surface [59,60,61,62]. TcMUC genes ensure the attachment to mammalian cells and immune system evasion and are subclassified into three groups according to their sequence, structure, and the parasite stage they are expressed. For example, TcMUC members are more abundant in amastigotes, while TcMUC II members are predominant in bloodstream trypomastigotes [57]. Like TSs, the mucins present high diversity that leads to an extensive repertoire of proteins composing the mosaic surface coat in the interplay between parasite and host, suggesting possible strategies to evade the immune system [30,63]. 

MASP genes were first identified during the *T. cruzi* genome sequencing and named due to their position among TS and mucin gene groups [28,64]. These proteins are up-regulated in mammalian-dwelling stages and are overexpressed in the infective forms [33]. They play a critical role in the invasion processing and survival of intracellular amastigotes [65]. 

Considering the crucial role of surface proteins as significant virulence factors in parasite infection, their current known regulatory mechanisms will be addressed in the following sections, emphasizing the mRNA signals, regulatory factors, and protein complexes acting on the gene expression modulation involved in *T. cruzi* differentiation.

## 2. Gene Expression Regulation of Significant Virulence Factors

Regulation of gene expression involves an extensive number of mechanisms that cells use to expand or reduce the production of determined gene products (protein or RNA). Virtually every stage of gene expression can be modulated: signal transduction, chromatin, chromatin remodeling, chromatin domains, transcription, post-transcriptional regulation, RNA transport, translation, and mRNA degradation. 

In most eukaryotes, the transcription initiation is the main point where gene expression regulation occurs [66]. However, trypanosomatids have developed strategies that differentiate them from other eukaryotes. The mechanisms related to the gene regulation in *T. cruzi* include alterations in gene expression, mainly at the post-transcriptional level, since its genes are transcribed as long polycistronic units produced by RNA polymerase II and then processed into single and mature transcripts [67,68]. 

The mRNA maturation process in trypanosomatids is known as splice leader trans-splicing (SL trans-splicing). After the polycistronic unit is transcribed, the processing involves the addition of the splice leader RNA at the 5′ end and the polyadenylation at the 3′ end. These short SL RNA molecules are divided in two regions, of which the 5′ half consists of the leader sequence transferred to a pre-mRNA and the SL RNA’s methylguanosine cap. The SL addition is required for mRNA stability, transport, and translation [69,70].

In the post-transcriptional steps of gene expression regulation, RNA localization is dictated by both cis-regulatory elements within the RNA molecule and trans-acting factors that can interact with the RNA. The cis-elements are recognized by specific trans-acting RNA-binding proteins (RBPs) that will form ribonucleoprotein (RNP) complexes. These RNPs target the RNA to specific cell regions, leading to protection from degradation, translation, or translation repression [71]. 

RBPs are trans-elements that are distinguished according to the RNA-binding domains (RBDs) they carry. They represent a large and well-conserved protein family, and the conservation usually is localized within the RBDs. These proteins often bind to RNA and other proteins known as accessory factors. RBPs can associate with specific RNA targets and recruit essential cofactors to arrange post-transcriptional gene regulation, including RNA maturation, nuclear export, stability, localization, and translation [72,73]. 

The cis-acting motifs can be called “zipcodes” since they direct mRNAs to their specific locations in the cell, and these regions can vary from a few nucleotides to around 1 kb in length [74]. Zipcode motifs are typically located in 3′ untranslated regions (UTRs), although, in contrast to transported RNAs in metazoans, they can also be found in coding regions in budding yeasts [75].

The protozoan parasites have a highly phenotypical heterogeneity to survive and achieve their life cycle through several conditions, including host immune pressure, drug pressure, vector, and host metabolic conditions to nutritional requirements. For these reasons, their cell surfaces are critical in protecting and detecting all these changes, and therefore protozoan parasites change their surface molecules constantly to adapt to environmental conditions. Moreover, these surface molecular variants are part of multigene families of surface proteins, which different paralogs can replace according to the requirements of the cell (e.g., vector or host) [76,77].

Thus, the stage-regulated gene expression is responsible for adjustments in the morphology, proliferation, and metabolism that distinguish the different forms of *T. cruzi* during its life cycle: the replicative intracellular amastigote, the infective bloodstream trypomastigote, the replicative epimastigote, and the infective metacyclic trypomastigote [10,17,78]. 

Considering the prevalence of mechanisms affecting post-transcriptional steps of gene expression in *T. cruzi*, RBPs are the main factors during the differentiation of protozoan parasites, interacting with specific regions within the mRNAs to regulate the expression of specific transcripts. In the following sections, we will describe the RBPs and mRNA regions involved in regulating the major surface molecules and other stage-specific proteins relevant to the virulence and survival of the parasite along its life cycle. 

### 2.1. Trans-Sialidases

In *T. cruzi*, the U-rich RNA binding protein (TcUBP-1) was identified as the primary regulator of this family of surface proteins. TcUBP1 is responsible for stabilizing/destabilizing many mRNAs, depending on the binding of other trans factors, through the parasite’s life cycle. These mRNP complexes contain members of the surface glycoprotein trans-sialidases (TcS) superfamily, mainly the ones expressed exclusively in infective trypomastigotes. Furthermore, TcUBP1 interacts with stage-specific transcripts containing SGPm (surface glycoprotein) next to the RNA structural element UBP1m, indicating that TcUBP1 can control the TcS mRNA expression in parasites going through differentiation to infective forms where these proteins are necessary. On the other hand, when TcUBP1 is overexpressed, the parasites can express the trans-sialidase enzyme, leading to metacyclogenesis [79].

### 2.2. Mucin and Amastin

A 241 nt cis-element of 241 nt associated with 3′ UTRs of the mucin family was also identified as part of 3′ UTR of trans-sialidases, MASP, mucin, and GP63. It seems to play an important role in the expression regulation of these multigenic families [80]. Moreover, there is evidence of a correlation between the existence of this repeat in the 3′ UTR of multigenic family genes and the levels of differential expression of these genes when comparing epimastigote and trypomastigote transcriptomes [81].

TcUBP1 also interacts with the AU-rich element located in the 3′UTR of mucin SMUG mRNAs, playing a role in the stabilization of the molecule [82]. The TcUBP-2 protein can act in transcript stabilization from the same family by binding to mRNAs’ poly(U) regions and is differentially expressed during parasite development. Both proteins interact in the same RNP complex and are implicated in controlling *T. cruzi* SMUG mucin mRNA levels [81]. 

Amastin is another type of surface glycoprotein that is up-regulated in amastigotes. It was first described in 1994 and has become important in the search for vaccine candidates for Chagas disease. Amastins were also characterized as large gene families in the genomes of several species of *Leishmania* and other trypanosomatids [83]. They have been described as proteins mainly involved in the parasite host–cell interaction and playing a part in the level of strain infectivity [84,85].

In 2000, Coughlin et al. reported a 68-fold difference in amastin mRNA levels between amastigotes and epimastigotes [86]. They also showed that amastin mRNA is seven times more stable in amastigotes than in epimastigotes, likely due to differences in the mRNA stability. In addition, it was shown that this cis-element must be positioned in a region between 250 nt from the stop codon and 180 nt from the polyadenylation site to confer its positive regulatory properties [86]. 

TcAlba30 was identified as the RBP of *T. cruzi*, acting as a negative regulatory factor in controlling β-amastin expression through interactions with its 3′UTR. Its overexpression in epimastigotes results in 50% decreased levels of β-amastin mRNAs compared to wild-type parasites [87].

### 2.3. GP82/GP90

GP82 is a surface glycoprotein, a member of the trans-sialidase family, expressed in metacyclic trypomastigotes. It is associated with parasite attachment to enter mammalian cells [88]. 

A study described elements in the 3′UTR of the GP82 mRNA involved in its regulation, leading to higher protein expression in metacyclic trypomastigotes than in epimastigotes. Cycloheximide treatment showed that GP82 mRNA is stable in metacyclic trypomastigotes, while the half-life of GP82 mRNA was dramatically increased in epimastigotes. This could be due to a mechanism mediated by a labile protein promoting the degradation and removal of stable GP82 mRNAs in epimastigotes but not in metacyclic trypomastigotes [89]. Bayer-Santos et al. generated a sequence of progressive deletions in the 3′UTR of GP82. Their results suggest that the mechanism regulating GP82 expression involves multiple elements in the 3′UTR [90]. 

GP90 is a metacyclic stage-specific surface glycoprotein, which affects the parasite’s ability to invade target cells. Contrary to the interaction mediated by GP82, GP90 binds to mammalian cells in a receptor-mediated manner without triggering a Ca2+ signal, which induces Ca2+ mobilization in both the parasite and the target cell [91].

Studies have suggested that the infectivity of *T. cruzi* metacyclic trypomastigotes is downregulated by the surface glycoprotein GP90. Gene expression regulation of GP90 starts early in the differentiation process, where different secretion pathways manage the delivery of the glycoprotein toward the cell surface. The mRNAs encode an N-terminal signal peptide and a C-terminal signal anchor driving polypeptide through the ER-Golgi secretory pathway to be glycosylated and receive a GPI anchor. GP90 proteins leave the Golgi complex in secretory vesicles that fuse with the flagellar pocket membrane and are distributed along the cell plasma membrane [91].

## 3. The Importance of RNP Granules in Modulating Gene Expression during *T. cruzi* Development

Regulation of gene expression in trypanosomatids occurs mainly at the post-translational level since mRNAs are transcribed into polycistronic units [92] and are further processed to generate mature mRNAs [17,93,94,95,96]. Access by mature mRNAs to the translation machinery needs to be accurately controlled for the correct progression through the different stages of parasite development. The mRNAs that are not undergoing translation accumulate in microscopically visible cytoplasmic RNP complexes formed by numerous proteins and RNAs called RNP granules [97,98,99,100]. Trypanosomatids have a vast repertoire of RNP granules, possibly an adaptation to the loss of transcriptional control. Excellent reviews accurately describe the various types of RNP granules, the proteins involved, and the relationship of these structures during parasite development [101,102,103]. However, although the importance of translational control and control of mRNA stability in regulating gene expression is evident, the connection between RNP granules and the regulation during the *T. cruzi* life cycle remains unknown.

The RNP granules described in *T. cruzi* [104,105] contain similar components to yeast or mammalian processing bodies (P-bodies), e.g., TcDHH1, and TcXRNA proteins. These proteins are expressed through the parasite life cycle and are localized in cytoplasmic foci that vary in number according to nutritional stress conditions and cycloheximide/puromycin treatment [106,107]. TcDHH1 and TcXRNA co-localize only at the nuclear periphery under cellular stress conditions and when the mRNA processing is inhibited [107]. 

The protein content usually indicates the granule type, and one protein can localize in more than one [108]. Moreover, distinct granule types can be present within the same cell and are usually formed due to different stimuli [102,109]. P-bodies are constantly present in the cell and respond to several stress conditions by increasing in number and size. They are sites of storage and/or degradation of several transcripts formed by the presence of translation-repressor proteins and components of the mRNA degradation machinery [97,105,110]. Starvation stress granules are likely involved in mRNA storage rather than degradation because they contain polyadenylated mRNAs and are stable at prolonged transcription inhibition [105]. Therefore, RNP granules play a central role in regulating the expression of genes involved in parasite differentiation since nutritional stress is critical in triggering metacyclogenesis [13]. Several investigations have demonstrated the presence of different RBPs in RNPs granules in *T. cruzi*, both in parasites in the exponential growth phase and under nutritional stress, suggesting an essential role of RBPs in post-transcriptional regulation [73,111,112,113,114].

Here, we will summarize how RBPs identified in RNP granules may regulate the fate of mRNAs during *T. cruzi* development. For this, we focused on some RBPs whose target transcripts have been identified through ribonomic approaches or by RT-qPCR analyses.

TcUBP1 is the best-known RBP involved in the stage-specific regulation of mRNAs in *T. cruzi* and is considered the model RBP in trypanosomes [81,101,103,105,115,116]. As previously mentioned, TcUBP1 is involved in the selective stabilization or destabilization of target transcripts, mainly for cell-surface glycoproteins that are preferentially expressed in the infective stages of *T. cruzi* [79,81,117]. An elegant study by Cassola et al. (2015) has shown that the association of TcUBP1 with RNA can be modulated according to the growth conditions to which the parasite is exposed [116]. In parasites under high nutrient/low-density growth conditions, TcUBP1 RNP complexes are distributed in a speckled pattern throughout the cytoplasm and nucleus and are lax and permeable to mRNA degradation. Conversely, under starvation conditions, TcUBP1 is associated with poly(A) mRNA in condensed cytoplasmic mRNA granules, which protect most of the mRNAs expressed in metacyclic forms from degradation [105].

Acetylation lowers binding affinity (ALBA) domain proteins have a general role in the development program of several microorganisms and exhibit unusual functional plasticity [118]. In trypanosomatids, ALBA proteins have been studied mainly in *Trypanosoma brucei* and are primarily implicated in the post-transcriptional regulation of gene expression. All Tb ALBA proteins interact with translation machinery components and can be recruited to cytoplasmic starvation granules [119,120]. In *T. cruzi,* TcALBA30 is localized into cytoplasmic granules under starvation conditions [121,122]. As mentioned in the previous section, TcALBA30 controls the steady-state levels of beta-amastin mRNAs in epimastigotes through elements present in their 3′-UTR, and the TcALBA30 over-expression in epimastigotes, resulting in 50% decreased levels of amastin mRNAs compared to wild type parasites. Therefore, the authors postulated that TcAlba30 acts as a negative regulator of beta-amastin mRNAs in *T. cruzi* epimastigotes [87]. 

TcDHH1 is a crucial component of RNP granules in trypanosomatids and is possibly involved in the remodeling of RNP complexes during parasite development. In general, mRNAs associated with TcDHH1 are not translated into proteins in epimastigotes but are predominantly translated during other stages of the parasitic life cycle [106]. Most of the mRNAs associated with TcDHH1 in epimastigotes are transcribed from MASP and mucin gene families. Additionally, the mRNA encoding amastin, a protein predominantly expressed in amastigotes, is also present in the TcDHH1 granules from epimastigotes [86]. Although the association of TcDHH1 with developmentally regulated mRNAs is evident, the exact role of this protein in regulating gene expression during the life cycle is still unknown, as is whether the granules that contain this protein participate in the storage or degradation of their target mRNAs. 

Dallagiovanna et al. (2008) demonstrated the association of TcDHH1 with TcPUF6 in the same RNP complex in epimastigotes. The Pumilio/FBF1 (PUF) family recognizes motifs in mRNA 3′-UTR and usually acts as a post-transcriptional repressor [123,124,125]. In *T. cruzi*, TcPUF6 seems to function as a repressor in epimastigotes because its mRNA targets are up-regulated in the infective metacyclic forms. Interestingly, TcPUF6 and TcDHH1 do not co-localize in metacyclic trypomastigotes, raising the possibility that TcPUF6 may depend on DHH1 for the destabilization of its target transcripts as part of the life cycle-specific regulation of gene expression [125].

The zinc finger protein ZC3H39 plays an essential role during the stress response in *T. cruzi*. Alves et al. (2014) have shown that TcZC3H39 is part of a protein complex that contains proteins to stress granules such as ribosomes, translation factors, TcDHH1, and other RBPs [126]. The mRNA targets associated with TcZC3H39 in unstressed epimastigotes are not associated with this protein in stress conditions, and vice versa, indicating that the composition of the mRNA targets shifts depending on the physiological conditions of the cell. Under stress conditions, the TcZC3H39-RNP complex sequesters both highly expressed mRNAs and their associated ribosomes, slowing translation activity, thereby reinforcing the importance of this RBP in the cellular stress response and mRNA metabolism in *T. cruzi* [126].

Recently, Tavares et al. (2021) characterized a zinc finger protein implicated in the control of epimastigote-specific gene expression. TcZC3H12 is up-regulated in epimastigotes and accumulates in granular structures, indicating that it might be part of the RNPs involved with mRNA storage. Furthermore, TcZC3H12 may have a regulatory role involved in epimastigote growth and differentiation since KO parasites have decreased growth rates and increased metacyclogenesis. Transcriptome analyses comparing wild type and TcZC3H12 KOs revealed that TcZC3H12 positively regulates transcripts required for epimastigote proliferation and negatively regulates transcripts required for differentiation into metacyclic trypomastigotes [127]. 

TcRBP9 is a developmentally regulated RNA-binding protein detected throughout the cytoplasm in epimastigotes but not in metacyclic trypomastigotes [128]. Proteomic analysis of the complexes containing TcRBP9 in epimastigotes and epimastigotes under nutritional stress demonstrated that TcRBP9 associates with distinct proteins in both cell growth conditions. In exponentially growing epimastigotes, TcRBP9 is associated with proteins known to be involved in mRNA metabolism, including TcDHH1 [104], TcALBA3/4 [119,120], TcZC3H39 [126], TcUBP1/2 [82], DRBD3 [129], and PABP1 [130]. However, when the parasite is subjected to nutritional stress, in addition to proteins involved in RNA metabolism, TcRBP9 is associated with translation initiation factors and PABP1, suggesting the participation of TcRBP9 in the stress granules formation [128]. Wippel et al. also performed RNA sequencing of isolated RBP9-RNP complexes and showed that most of these transcripts mapped to hypothetical proteins. However, some transcripts identified encode RBPs and proteins involved in metabolic processes. Furthermore, when they compared their data with the ribosome profiling data from Smircich et al. 2015, they observed that all these transcripts were identified in exponential growth epimastigotes. Given that TcRBP9 was not detected in trypomastigotes, these data suggest a positive translational regulation of these transcripts in epimastigotes by the RBP9-RNP complex [39].

These studies provide insight into how RBP-RNA interplay can modulate gene expression. The dynamic association of different RBPs in RNP complexes seems to regulate the assembly of RNA granules and the fate of stage-specific transcripts. During low nutrient availability in the epimastigote stage (insect vector), mRNA granules seem to be important. Given that nutritional stress is essential for triggering differentiation, we conclude that RNP granules play an important role in modulating the gene expression of distinct mRNA, including the one coding for virulence factors during *T. cruzi* development.

## 4. Conclusions

The protozoan *T. cruzi* is the causative agent of Chagas disease, a neglected tropical disease. Therefore, it is essential to understand the fundamental molecular and cell biology involved in the pathogenesis of this parasite. Surface proteins play a crucial role as significant virulence factors in parasite infection since they confer the main proprieties for infectivity and survival inside the hosts. Several proteins are regulated during the parasite cycle to confer the biological properties of each specific stage. The coat remodeling in the infectivity form is critical to guarantee the host cell infection, protein adhesion, cell signaling, and mammalian host immune response modulation. The development of high-throughput methodologies has increased the identification of stage-specific genes and provided a broad panorama of the major factors involved in parasite differentiation. Surface proteins are among the most abundant and significant proteins in this process. In addition to studies aimed at understanding their role in parasite infection, several studies focused on investigating the factors responsible for modulating their expression. Unlike other eukaryotes, trypanosomes control gene expression mainly in the post-transcriptional process due to the peculiarities of their gene expression, in which different genes are transcribed into the same polycistronic RNA. Thus, the mechanisms that affect mRNA stability and translation are the most described so far. Distinct RBPs and mRNA cis-elements have been reported, mainly for significant surface proteins involved in virulence processes.

Although different RBPs have been identified, TcUBP1 emerged as a good model of RBP in *T. cruzi* since its mode of action is well known for controlling the prominent surface protein families, such as TS and mucins. Additionally, the investigation of TcUBP1 and other RBPs highlighted that parasites’ RBPs co-regulate functionally related mRNAs mainly at stability and translation. Genome-wide methods identified subsets of functionally related mRNAs that associate with RBPs coordinating expression of mRNAs sharing the same *cis*-elements. This seems to be the case for the significant surface families of *T. cruzi*. Some of these mRNAs undergo simultaneous decay, whereas some are translationally co-regulated by polysomes. There is considerable evidence that RNP granules control the gene expression in this parasite, although the composition is not very similar to the granules described in other eukaryotes. RNP granules seem to share proteins common to other eukaryotes and respond to stimuli, mainly nutritional stress, during the differentiation in infective forms. However, the precise mechanism and how they respond to stimuli remain unknown.

## Data Availability

Not applicable.

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
