# Peer review of "Modulation of Virulence Factors during Trypanosoma cruzi Differentiation"

_pathogens, 2022, doi:10.3390/pathogens12010032_

Round 1
Reviewer 1 Report
In this manuscript authors review modulation of virulence factors during differentiation of Trypanosoma cruzi, and the important role of RNA Binding Proteins (RBP) regulating the gene expression of these factors in different life stages of the parasite.
Overall, the manuscript reveals important details about molecular mechanisms of post-transcriptional and post-translational regulation in T. cruzi, and provide a fundamental information about surface proteins involved in differentiation process and infectivity, like Trans-sialidases, Mucins, MASPs and Amastins. Authors address the role of RBPs in gene expression regulation of these proteins and analyze how different the ribonucleoprotein (RNP) granules modulate gene expression during T. cruzi development. While manuscript is well written, however, it requires minor revisions.
There are some minor comments that should be assessed:
· There are some errors in citations that could be related to the bibliographical citation manager used:
- Check lines 33, 227, 230, 232 and 306, cites appear with complete names of authors, instead numbers.
- Check scientific names in references, since all of them should be in italic and epithets should start with lowercase.
- There is not the cite and reference of Atwood et al. 2005 (Line 80-85)
· It is necessary to name proteins in one style. For example, GP63 protein (Line 109) is also written as Gp63 (Line 124); TcDHH1 (Line 354) and TcDhh1 (Line 359)
· Check the spaces between words, some places there seem to be double spaces.
· Line 37: The order of parasites names could be improved: Trypanosoma cruzi, T. brucei and Leishmania sp (for instance)
· Line 39: Words “triatomines from the” can be omitted since all triatomines are from family Reduviidae.
· Lines 43-48: Localization of every parasite stage at invertebrate vector and mammalian host and how it is transmitted should be clear. For instance, the differentiation from epimastigotes to metacyclic trypomastigotes occurs in hindgut (rectum) of vector. The mammalian host can get infected when bug feces enter the body through mucous membranes or breaks in the skin.
· Line 68: References described in this cite belong to 1994 to 1999, so they should not be called as studies from the 2000s. It should be replaced by other time frame, like “last decade of XX century”
· Lines 75-77: There are many “of” connectors (5). The phrase could be reorganized to avoid this.
· Line 214: It is better to include the full name of the protein: U-rich RNA Binding Protein 1 (TcUBP-1).
· Lines 224-252: I do not see any information about RBP regulating MASPs in the paragraph, although it is related in section title
· Line 225: Should be a space between 241 and nt
· Line 240: Leishmania should be in italic.
· Line 296: … the parasite life cycle e are localized in… The e is misplaced
· Line 339: It should be 3’ UTR
· Line 342: The word "beta" must be joined
· Line 356: The meaning of UTR must be where the term UTR is used for the first time in the manuscript, however, it is named last.
· Line 364: …contains proteins to stress granules such as… The to is misplaced
As final comments, the review fits the journal’s scope, it has a good English quality and organization. I recommend this manuscript for publication with minor revisions expressed above.
Author Response
REVIEWER 1
"In this manuscript authors review modulation of virulence factors during differentiation of Trypanosoma cruzi, and the important role of RNA Binding Proteins (RBP) regulating the gene expression of these factors in different life stages of the parasite.
Overall, the manuscript reveals important details about molecular mechanisms of post-transcriptional and post-translational regulation in T. cruzi, and provide a fundamental information about surface proteins involved in differentiation process and infectivity, like Trans-sialidases, Mucins, MASPs and Amastins. Authors address the role of RBPs in gene expression regulation of these proteins and analyze how different the ribonucleoprotein (RNP) granules modulate gene expression during T. cruzi development. While manuscript is well written, however, it requires minor revisions."
Authors: we appreciate the comments, and we revised all the manuscript addressing the points below as suggested. The additional or adjusted information are in red to simplify since the line numbers changed due addition of information. The track changes (in blue) refer to English edition of the text. Bellow, see our answers, point by point, in bold and italic.
There are some minor comments that should be assessed:
- There are some errors in citations that could be related to the bibliographical citation manager used:
- Check lines 33, 227, 230, 232 and 306, cites appear with complete names of authors, instead numbers.
Authors: we have adjusted them. Now the citations are in numbers.
- Check scientific names in references, since all of them should be in italic and epithets should start with lowercase.
Authors: we have adjusted them.
- There is not the cite and reference of Atwood et al. 2005 (Line 80-85).
Authors: we have included it.
- It is necessary to name proteins in one style. For example, GP63 protein (Line 109) is also written as Gp63 (Line 124); TcDHH1 (Line 354) and TcDhh1 (Line 359).
Authors: we have adjusted them.
- Check the spaces between words, some places there seem to be double spaces.
Authors: we have checked it.
- Line 37: The order of parasites names could be improved: Trypanosoma cruzi, T. brucei and Leishmania sp (for instance).
Authors: we have adjusted them.
- Line 39: Words “triatomines from the” can be omitted since all triatomines are from family Reduviidae.
Authors: we have adjusted them.
- Lines 43-48: Localization of every parasite stage at invertebrate vector and mammalian host and how it is transmitted should be clear. For instance, the differentiation from epimastigotes to metacyclic trypomastigotes occurs in hindgut (rectum) of vector. The mammalian host can get infected when bug feces enter the body through mucous membranes or breaks in the skin.
Authors: we rephrased it as suggested.
- Line 68: References described in this cite belong to 1994 to 1999, so they should not be called as studies from the 2000s. It should be replaced by other time frame, like “last decade of XX century”.
Authors: we have adjusted it.
- Lines 75-77:There are many “of” connectors (5). The phrase could be reorganized to avoid this.
Authors: we have adjusted them.
- Line 214: It is better to include the full name of the protein: U-rich RNA Binding Protein 1 (TcUBP-1).
Authors: we have adjusted them.
- Lines 224-252: I do not see any information about RBP regulating MASPs in the paragraph, although it is related in section title.
Authors: we have adjusted the title.
- Line 225: Should be a space between 241 and nt
Authors: we have adjusted it.
- Line 240: Leishmania should be in italic.
Authors: we have adjusted it.
- Line 296: … the parasite life cycle e are localized in… The e is misplaced
Authors: we have adjusted it.
- Line 339: It should be 3’ UTR
Authors: we have adjusted it.
- Line 342: The word "beta" must be joined
Authors: we have adjusted it.
- Line 356: The meaning of UTR must be where the term UTR is used for the first time in the manuscript, however, it is named last.
Authors: we have adjusted it.
- Line 364: …contains proteins to stress granules such as… The to is misplaced
Authors: we have adjusted it.
As final comments, the review fits the journal’s scope, it has good English quality and organization. I recommend this manuscript for publication with minor revisions expressed above.
Authors: we really appreciate your words, and we improved the manuscript for the final version. Hope it is in accordance now.

Reviewer 2 Report
The manuscript "Modulation of virulence factors during Trypanosoma cruzi differentiation" summarizes the knowledge about the virulence factors in T. cruzi, and the potential post-transcriptional regulators during its complex life cycle across the different parasite stages.
The manuscript can positively contribute to the understanding of the virulence factors in this parasite. However, some major changes and/or updates should be made prior to its publication.
- First, section number 1 is not concise. It's hard to understand whether the authors are trying to focus on proteomics, transcriptomics, surface proteins, or everything at the same time. The authors present the results from some articles, but the discussion as a joint is scarce. For example, paragraphs in lines 95-127 are unconnected or not discussed at all.
- Section 2. The importance of the splice leader sequence (SL) during transcription and gene expression must be included and discussed. It is not even mentioned at all.
- Statements about specific results or conclusions should refer to the original work where the finding was made, rather than making reference to the reviews where the finding is just mentioned. It is a systematic problem in the whole manuscript. For example the statements in lines 281-283 and 300-302.
- Some statements making or proposing systematic expression profiles must be complemented with all the references available (the most recent if possible), rather than referring to a single manuscript, for example, those in lines:
89-91, 106-107, 123-125, and 151-152
- Lines 72-74. To this date, there are 38 entries in the NCBI.
General comments and minor typos.
- Authors should avoid over-citing reviews more than 5 years old. Discussing the most recent publications should be preferred.
- Typos in line 296
- 393-396, Do the authors mean "their work" instead of "this work"?
All in all, the objective of a review is to summarize the most relevant and recent findings in the field, in order to keep the scientific community updated. In consequence, the authors should focus on the most recent manuscripts that support old/new hypotheses or describe new findings.
Many recent manuscripts about transcriptomics and proteomics during the T. cruzi life stages, and/or describing the mRNA processing and gene expression control were not included in this review.
Author Response
REVIEWER 2
"The manuscript "Modulation of virulence factors during Trypanosoma cruzi differentiation" summarizes the knowledge about the virulence factors in T. cruzi, and the potential post-transcriptional regulators during its complex life cycle across the different parasite stages.
The manuscript can positively contribute to the understanding of the virulence factors in this parasite. However, some major changes and/or updates should be made prior to its publication.
Authors: we appreciate the comments, and we revised all the manuscript addressing the points below as suggested. The additional or adjusted information are in red to simplify since the line numbers changed due addition of information. The track changes (in blue) refer to English edition of the text. Bellow, see our answers, point by point, in bold and italic.
- First, section number 1 is not concise. It's hard to understand whether the authors are trying to focus on proteomics, transcriptomics, surface proteins, or everything at the same time. The authors present the results from some articles, but the discussion as a joint is scarce. For example, paragraphs in lines 95-127 are unconnected or not discussed at all.
Authors: Thanks to highlight this point, and we would like to clarify first that the idea of the first section would give an overview and somehow a temporal timeline of the studies involving the identification of gene-specific proteins and genes during the differentiation of infective forms. We would like to describe the different methods and approaches to identify stage regulated genes, mainly using the differentiation in vitro method since it has been widely used for this purpose. Indeed, we would like to give an overview how the methodologies improved the knowledge in the area, highlighting that surface proteins are the major stage specific virulence factors. Thus, the other sections we provide the recent findings of regulation mechanisms and factors responsible by the regulation of those proteins, the focus of the revision. Since this idea was not clear, we modified different parts of this section, adding information, and improving the connection of the ideas. We hope that the purpose of this section is more appropriated now.
- Section 2. The importance of the splice leader sequence (SL) during transcription and gene expression must be included and discussed. It is not even mentioned at all.
Authors: we have adjusted it.
- Statements about specific results or conclusions should refer to the original work where the finding was made, rather than making reference to the reviews where the finding is just mentioned. It is a systematic problem in the whole manuscript. For example, the statements in lines 281-283 and 300-302.
Authors: We understand it and agree that original works are the most appropriate. We have tried to use the original ones as much as possible, but the data on surface proteins have been obtained since 80's years and in this case we thought it would simplify to use some good quality and update reviewers to give the opportunity to the readers to access more detailed information regarding the topic. Anyway, we checked some statements and included some additional works to improve the manuscript.
- Some statements making or proposing systematic expression profiles must be complemented with all the references available (the most recent if possible), rather than referring to a single manuscript, for example, those in lines:
89-91, 106-107, 123-125, and 151-152
- Lines 72-74. To this date, there are 38 entries in the NCBI.
Authors: as we explained above, we chose to simplify the access to the data citing other revisions since the focus of these review is gene expression regulation and not the function of virulent factors. We also aware that there several papers on description of stage-specific genes in different stages of parasite, such as amastigote and amastigotes but we decided to keep the focus on use the ones related to differentiation into infective forms. Anyway, we revised some statements and include additional papers.
General comments and minor typos.
- Authors should avoid over-citing reviews more than 5 years old. Discussing the most recent publications should be preferred.
Authors: we have improved it
- Typos in line 296
Authors: we have adjusted it.
- 393-396, Do the authors mean "their work" instead of "this work"?
Authors: we have adjusted it.
All in all, the objective of a review is to summarize the most relevant and recent findings in the field, in order to keep the scientific community updated. In consequence, the authors should focus on the most recent manuscripts that support old/new hypotheses or describe new findings.
Many recent manuscripts about transcriptomics and proteomics during the T. cruzi life stages, and/or describing the mRNA processing and gene expression control were not included in this review.
Authors: we improved it, including additional papers ( numbers of references in red). However, as we explained above, we also would like to give an overview and describe how the studies developed along the years, focusing on differentiation of infective forms, mainly on surface proteins instead of include data from other stages and genes and regulatory elements. Hope that our intention is clear now and the changes made could improve the manuscript as suggested.

Round 2
Reviewer 2 Report
An error in lines 199-201 should be corrected before the publication of the manuscript. Genes are not expressed as long polycistronic transcripts, they are transcribed as long polycistronic units, then processed into single and mature transcripts.
Author Response
We adjusted lines 199-201 as recommended by the reviewer.